# Detection, Isolation and Quantification of Myocardial Infarct with Four Different Histological Staining Techniques

**DOI:** 10.3390/diagnostics14202325

**Published:** 2024-10-18

**Authors:** Xiaobo Wu, Linnea Meier, Tom X. Liu, Stefano Toldo, Steven Poelzing, Robert G. Gourdie

**Affiliations:** 1Center for Heart and Reparative Medicine Research, Fralin Biomedical Research Institute at Virginia Tech Carilion, Roanoke, VA 24016, USA; xbwu@vt.edu (X.W.);; 2Department of Biological Sciences, Virginia Tech, Blacksburg, VA 24061, USA; 3Center for Artificial Intelligence, Bluhm Cardiovascular Institute, Northwestern University, Chicago, IL 60201, USA; 4Robert M. Berne Cardiovascular Research Center, Department of Medicine, Division of Cardiovascular Medicine, University of Virginia, Charlottesville, VA 22908, USA; 5Department of Biomedical Engineering and Mechanics, Virginia Tech, Blacksburg, VA 24061, USA

**Keywords:** myocardial infarction, quantification, MATLAB, color-based, staining techniques, fibrosis

## Abstract

Background/Objectives: The precise quantification of myocardial infarction is crucial for evaluating therapeutic strategies. We developed a robust, color-based semi-automatic algorithm capable of infarct region detection, isolation and quantification with four different histological staining techniques, and of the isolation and quantification of diffuse fibrosis in the heart. Methods: Our method is developed based on the color difference in the infarct and non-infarct regions after histological staining. Mouse cardiac tissues stained with Masson’s trichrome (MTS), hematoxylin and eosin (H&E), 2,3,5-Triphenyltetrazolium chloride and picrosirius red were included to demonstrate the performance of our method. Results: We demonstrate that our algorithm can effectively identify and produce a clear visualization of infarct tissue in the four staining techniques. Notably, the infarct region on an H&E-stained tissue section can be clearly visualized after processing. The MATLAB-based program we developed holds promise for infarct quantification. Additionally, our program can isolate and quantify diffuse fibrotic elements from an MTS-stained cardiac section, which suggests the algorithm’s potential for evaluating pathological cardiac fibrosis in diseased cardiac tissues. Conclusions: We demonstrate that this color-based algorithm is capable of accurately identifying, isolating and quantifying cardiac infarct regions with different staining techniques, as well as diffuse and patchy fibrosis in MTS-stained cardiac tissues.

## 1. Introduction

Myocardial infarction is one of the leading causes of cardiac death in the world [1,2]. The infarct size is highly associated with the incidence of cardiac arrest [3,4]. Therapeutic strategies have been developed to reduce the infarct size and prevent the incidence of cardiac arrhythmias [5,6,7,8]. Therefore, improving the accuracy of the quantification of the infarct region is imperative to assess the efficacy of such therapeutics.

Currently, manually tracing the infarct region with the aid of image analysis software remains the conventional method of quantifying myocardial infarct size [9,10,11]. However, this hand-tracing method can pose challenges that may impact the accuracy, cost and expediency associated with image analysis. Despite recent developments in algorithm-based methods, poor user-friendliness and a limited histological staining technique discretion hinder the potential use of such methods [12,13,14,15]. 

Masson’s trichrome (MTS) and picrosirius red (PSR) are most commonly used to detect and assess the post-myocardial infarction scar, which is easily distinguishable as it is blue for MTS and red for PSR, relative to the non-infarct region [16,17]. Given this distinct color difference, we aimed to develop an easy and effective algorithm to quantify the myocardial infarction with different histological staining techniques.

In comparison with MTS, PSR and 2,3,5–Triphenyl tetrazolium chloride (TTC) [6], hematoxylin and eosin (H&E) is rarely used for infarction evaluation and is sometimes used to understand the morphology of the infarct without isolating and quantifying the infarct region [18,19]. In this study, H&E staining was also included to investigate whether the infarct region can be clearly identified and isolated with our algorithm. Additionally, we evaluated whether our algorithm is capable of isolating and quantifying diffuse fibrosis. 

## 2. Methods

The investigation conforms to the Guide for the Care and Use of Laboratory Animals published by the US National Institutes of Health (NIH Publication No. 85–23, revised 1996). All animal study protocols were approved by the Institutional Animal Care and Use Committee at the Virginia Polytechnic Institute and State University.

### 2.1. Histological Staining

Masson’s trichrome staining (MTS): The 8–12-week-old CD-1 mouse hearts on which we performed the cryoinjury method, which served as a model for myocardial infarction [8], were fixed in 4% paraformaldehyde for 48 h. Paraffin-embedded sections with 5 µm thickness were taken from the middle of the scars and then stained with a Masson’s Trichrome Kit (Poly Scientific, Bay Shore, NY, USA) according to the protocol provided by the manufacturer.

Hematoxylin and eosin staining (H&E): The 8–12-week-old CD-1 mouse hearts that underwent cryoinjury were fixed in 4% paraformaldehyde for 48 h, followed by paraffin embedding. The entire heart was sectioned into 5 µm thick slices, with 10 µm intervals between each section. All sections were then stained with H&E (Sigma-Aldrich, St. Louis, MO, USA) [8].

Triphenyl-tetrazolium-chloride staining (TTC): The mice were euthanized 24 h after the surgical induction of ischemia/reperfusion injury. The hearts were collected and connected to an antegrade perfusion apparatus. After the blood was washed out, 10% triphenyl tetrazolium chloride (Sigma-Aldrich) in PBS 1X was infused. The coronary artery was reoccluded and the myocardium was infused with a PBS 1X solution containing 1% Phthalo blue dye (Quantum Ink, Louisville, KY, USA) and 5 mM adenosine. Afterward, the hearts were removed, frozen and cut into transverse slices of equal thickness, about 1 mm, from apex to base [20].

Picrosirius red (PSR): The mice were euthanized 7 days after the surgical induction of ischemia/reperfusion injury. The hearts were immersed in 10% formalin for 48 h, embedded with paraffin and sliced into 5 μm thick slices. Tissue slides were stained with picrosirius red using a staining kit, following the manufacturer’s instructions (Poly Scientific R&D Corp., Bay Shore, NY, USA) [21].

### 2.2. Pre-Processing of Stained Tissues 

All stained tissue sections were scanned and stored as color digital images which consist of a combination of three primary colors: red, green and blue (RGB), and then employed in MATLAB software (MATLAB R2022b, MathWorks, Natick, MA, USA) for image analysis.

### 2.3. Rationale

Relative to other techniques, MTS is predominately used for assessing myocardial infarction or fibrosis. With MTS, the infarct region is distinguishable by an apparent blue color relative to other non-infarct regions (Figure 1A, left). In the MATLAB program, we analyzed RGB color value proportions in both the infarct regions and non-infarct regions. As represented in Figure 1A, left, the R values in an infarct region (41) were considerably smaller than that in non-infarct regions (251). Notably, this difference can be utilized to exclude non-infarct regions by removing pixel values that exceed an R value limit using a color threshold. In order to improve the isolation accuracy of this mechanism, a ratio of B and R values was designed to support the extraction of normal tissues and any pixels that had very faint blue colors. As shown in Figure 1A, right, the blue infarct region was successfully isolated and the non-infarct region was removed or greatly suppressed.

Isolating the infarct region after TTC staining was relatively simple, as TTC stains normal tissues a deep red and renders the infarct region an off-white color (Figure 1B, left). The numeric quantities of R, G and B in the infarct region were comparable in value (133, 136 and 134, respectively), whereas the red value (187 vs. 69 and 61) in the non-infarct region was dominant. Due to the similarity between the values for R, G and B in the infarct region, a standard deviation threshold was employed to separate both infarct and non-infarct tissue. Unscarred regions contained pixels with various color values, which produced high standard deviations. The contrast between high and low standard deviations within the color code of differing locations of the tissue produced a clear differentiation between infarcted and non-infarcted structures, which gave rise to the successful isolation of the infarct region in TTC-stained sections (Figure 1B, right). 

Similar to the distinct color difference between the infarct region and non-infarct region in the TTC image, the infarct region in the PSR-stained section is stained as a dominant red color (240 vs. 129 and 143) relative to the non-infarct region (204, 191 and 174, respectively) (Figure 1C, left). In order to improve the isolation accuracy, we designed a new color-based algorithm to extract the infarct:$$mean_{GB}=\frac{G+B}{2}$$
$$R>mean_{GB}\;\mathrm{or}\;R-mean_{GB}\geq thd,$$

*G* and *B* represent green and blue values in a pixel, respectively; *mean_GB_* represents the mean values of *G* and *B*, and *R* represents the red value in a pixel. *thd* is a user-determined threshold for extracting the infarct. With the algorithms above, the infarct region can be successfully isolated from normal tissue (Figure 1C, right). 

To assess the algorithm’s potential to identify and isolate the infarct region in H&E-stained tissues, an H&E image was compared to an analogous MTS-stained tissue. The tissue section next to the section stained with MTS, labeled “Original” (Figure 2A, left), was stained with H&E (Figure 2B, left). As shown in Figure 2B, left, it is challenging to visualize the infarct region in the original H&E-stained tissue. However, one can roughly identify the infarct region by zooming in on the image to recognize the difference in the muscle fibers. After comparing RGB values between the infarct and non-infarct regions, it was observed that the value for green (174) in the infarct region is larger than that in the non-infarct region (99). However, it was not feasible to isolate the infarct region by utilizing the same color threshold as the MTS images. Therefore, we designed two steps to reveal the infarct. In the first step, a threshold was applied to remove large values of *R*, *G* or *B* in a pixel. If an *R* value was larger than the threshold, it would be converted to zero, but the *G* and *B* values in this pixel did not change. With this threshold, the infarct is visualized with the color green (Figure 2B, middle) and the normal tissues are changed to light blue. In the second step, only the green color indicated that the infarct was extracted, and any other colors were removed. Specifically, in a pixel, if the R value or *B* value is smaller than the *G* value, the *R* and *B* values will be changed to zero. Eventually, the infarct region is clearly identified with mildly suppressed normal tissue regions (Figure 2B, right). Later, the large and dense area of green in the infarct region can be masked for isolation and dilated for quantification, which will be described below. Overall, the comparison of highlighted infarcts after MTS and H&E could demonstrate that the algorithm could provide a clearer visualization of the infarct region with H&E staining. 

Overall, our color-based algorithm systematically identified and isolated the infarct region from the non-infarct region regardless of staining technique. Next, we developed a MATLAB-based user-friendly graphic program to isolate and quantify the infarct region.

## 3. Results

MATLAB has a powerful image processing toolbox, which was used to mask the infarct region and thereby quantify the infarct size. We develop here complete instructions regarding how to use our algorithm to analyze the infarct region. All necessary codes and instructions can be found in the Appendix A. 

### 3.1. Image Splitting

Our program analyzes one image at a time. Since several tissue sections are normally placed and stained on a slide, and then the entire slide may be scanned with a camera to create a digital image (Figure 3A), a necessary step is required to split all the section images. Here, we developed a simple MATLAB-based graphic interface named *split_image* to split them without altering the image resolution. This interface can split up to five single images (Figure 3B). The top and bottom of each image can be adjusted to make sure each sub-image can be split. After splitting, every single image can be saved on a local computer (Figure 3C) using a button.

### 3.2. Isolation and Masking of Infarct Area in the Section

We developed a MATLAB-based graphic program named *scar_size_analysis* to analyze the myocardial infarction in each image from the four different staining techniques. The related operational instructions can be found in the Appendix A. Briefly, an image is loaded into the program by selecting the staining method from MTS, H&E, TTC or PSR. The color threshold for detecting and isolating the infarct is specifically pre-defined for each staining technique, which can be adjusted by the user. The infarct region can be cropped for masking and quantification. With this program, the infarct region in each stained image can be detected, isolated and masked as shown in Figure 4. Notably, the infarct region in the H&E-stained tissue section can be detected and isolated clearly (Figure 4B). Subsequently, the isolated infarct can be dilated and masked with the white color for quantification. Notably, artifacts, such as the bubble in the center of the TTC section (Figure 4C), can be removed during masking, which presents user-determined pre-processing opportunities for improving quantification accuracy. Furthermore, the entire tissue section can also be masked. After masking, the total number of pixels in the infarct and surrounding non-infarct tissue regions comprising the remainder of the sample, a percentage of the infarct region relative to the whole section and the images produced during this analysis can be saved for further analysis. 

### 3.3. Quantitative Analysis of Infarct 

For infarct area quantification analysis, this program supports two quantitative parameters: the percentage of infarct region relative to the entire section and the actual area of the infarct region. Mathematically, the percentage of infarct region relative to the entire section can be calculated as follows:$$Relative\;infarct\;size\;=\;\frac{the\;number\;of\;pixels\;in\;the\;infarct\;region\times 100}{the\;number\;of\;pixels\;in\;the\;entire\;section}\%$$

The number of pixels in the infarct region and in the entire section are produced from the program *scar_size_analysis*.

To calculate the actual infarct area, the pixel size after scanning the image with a microscope is required and can be found in the log. It is commonly assumed that the pixel is a square. With the known pixel size, the actual infarct area can be calculated as follows:Infarct area = Px×Px×the number of pixels in the infarct region

*Px* represents the pixel size. The area of the entire section can also be determined by the number of pixels in the entire section:Section area =P x×Px×the number of pixels in the entire section

We also provide instructions to quantify the infarct volume in the entire heart if it is required. To quantify the infarct volume in a heart, the entire heart must be sectioned and stained. The thickness of each section must be considered for volume quantification. To accelerate processing, tissues with a fixed thickness between sections may be removed, and an assumption can be made that the infarct area and section area in the removed tissue are equal to that in the adjacent section. To quantify the infarct volume, the infarct area in each section is calculated first, which has been described above. Then, the infarct volume (Vs) in each section can be calculated as the infarct area (As) multiplied by the section height (h1). The assumed infarct volume in the removed section (Vrs) will be the infarct area (As) multiplied by the removed section height (h2). Once the infarct volumes in each section and the removed section are acquired, the sum of the infarct volumes from all sections will be the infarct volume of a heart. The detailed mathematical equations can be found in Figure 5. This instruction is also applicable to quantifying the tissue volume of the entire heart. With this method, we quantified the volumes of the infarct and entire ventricle from a male CD-1 mouse heart suffering from cryoinjury-induced myocardial infarction [8]. Specifically, over 80 sections with a thickness of 5 µm were stained with H&E, and the thickness of the removed tissue between two adjacent sections was 10 µm. After analysis, the volumes of the infarct region and the entire ventricles were 11.15 mm^3^ and 134.37 mm^3^, respectively. Therefore, the relative infarct volume in this heart was about 8.3% of the entire ventricles. Some representative in-series images with highlighted infarcts can be found in Figure 6.

### 3.4. Quantification of Cardiac Fibrosis

Increased patchy and diffuse cardiac fibrosis is often associated with cardiac disease and aging [22,23]. The pattern of this type of fibrosis makes it challenging to quantitatively evaluate fibrosis levels. Although the primary purpose of this algorithm is to evaluate myocardial infarction, the color difference between the fibrotic elements in cardiac tissue and normal tissue makes it possible to isolate fibrotic elements from the entire section and thereby exclusively investigate the distribution and pattern of fibrosis in the heart. In order to explore whether our algorithm is capable of detecting and isolating sparse fibrotic elements in a cardiac tissue section, we developed another program, named *fibrosis_analyzer*, focusing on diffuse fibrosis evaluation, separate from the program that is described above, to assess myocardial infarction. At this point, the program only analyzes cardiac fibrosis in an MTS-stained tissue section, though with the same algorithm as used for infarction analysis. To examine if the program can isolate and quantify diffuse fibrosis, cardiac tissues from wild-type and desmoplakin-deficient mice were used. The deletion of the desmoplakin gene is highly associated with increased cardiac fibrosis [24]. We found that a tissue section from a 10-week-old wild- type C57BL/6J mouse has 2.5% cardiac fibrosis (Figure 7A) relative to the 9.7% cardiac fibrosis observed in a 10-week-old desmoplakin-deficient mouse heart, which shows more visible fibrosis across the ventricles (Figure 7B). By contrast, cardiac fibrosis reaches 18.9% in a 22-week-old desmoplakin-deficient mouse heart, which suggests that cardiac fibrosis increases with aging (Figure 7C). Notably, the isolation of fibrosis also helps us visualize fibrosis distribution across the ventricle wall. In the isolated fibrosis images in Figure 7, we noticed that cardiac fibrosis is mainly distributed in the left ventricle, with very low levels of fibrosis in the septum and right ventricle, regardless of the genetic background or age. Overall, after fibrosis isolation, we obtained a general impression of the fibrosis distribution across the ventricular wall without the distraction of the overwhelming large non-fibrotic tissues.
Figure 5The schematic and equations for calculating the area and volume of the tissue and the infarct. A represents the area of the tissue (*A_t_*) or infarct (*A_s_*); h represents the height of the section (*h*1) or removed tissues (*h*2); *P_x_* represents the pixel size; ∑*P* represents the number of pixels in the infarct (∑*P_s_*) or the tissue (∑*P_t_*); *V_si_* and *V_ti_* represent the infarct and the tissue volume in each section, respectively; *V_rsi_* and *V_rti_* represent the infarct and the tissue volume in each removed tissue, respectively; and *V_Ts_* and *V_Tt_* represent the total volume of the infarct and the tissue in this heart, respectively. A1, A2…An represent the sequential order of all stained sections; h1 represents the thickness of each stained section; h2 represents the thickness of the removed tissues.
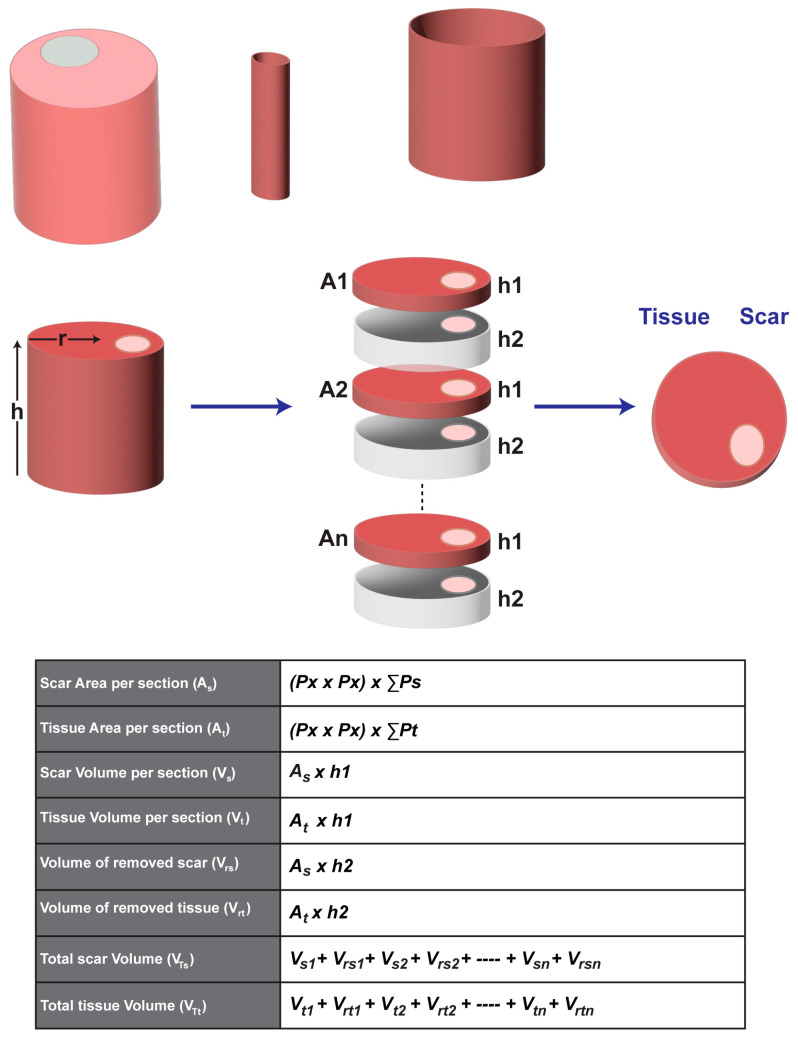

Figure 6Representative in-series images with detected myocardial infarcts in H&E-stained tissue sections in a heart. The images in the series show the gradually larger area of infarction followed by the diminishing area of infarction in subsequent sections. The percentage number in each image represents the percentage of the infarct in that section.
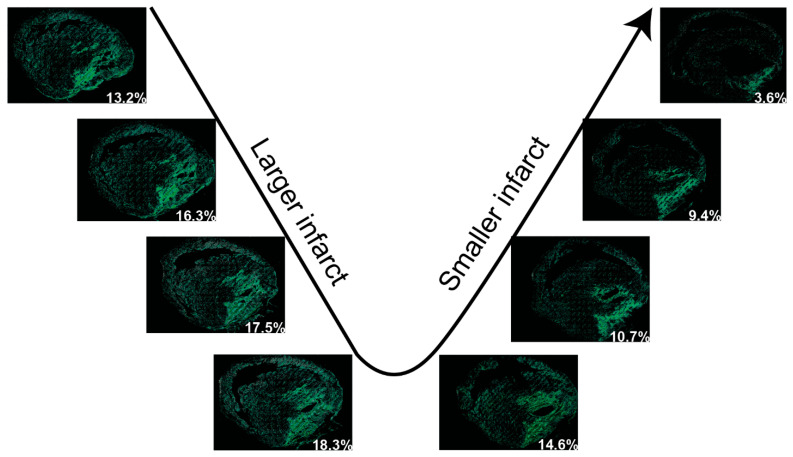


## 4. Discussion

Histological staining techniques primarily stain the myocardial infarct with one color and non-infarct tissues with another color, which provides the opportunity for isolating the infarct with the aid of advanced imaging microscopy and programming techniques. Multiple semi-automatic approaches have been developed to quantify the myocardial infarct, while they were only used to process the images from a single histological staining technique [12,13,14,15]. By contrast, the algorithm we developed in the current study can be used for infarct assessment in four different histological staining techniques. Although there are slight differences in image processing in different staining techniques, the rationale behind these different processing steps is the same—namely, we take advantage of differences in color coding between infarcted and non-infarcted regions, regardless of histological staining technique. Conveniently, this rationale is generally applicable for infarction assessment in different organs, including cerebral infarction induced by stroke [25,26]. Notably, the algorithm we provide also has the additional potential to allow for the quantification of diffuse fibrosis levels as well. Overall, our simple and effective algorithm demonstrated the capability of identifying, isolating and quantifying myocardial infarct in tissues stained with either MTS, H&E, TTC or PSR. 

It is well known that MTS staining is commonly used to label myocardial infarction or fibrosis with the color blue. Using this as a control/baseline, we successfully validated the detection and isolation of infarcts in H&E-stained tissue. The similarity in color between the infarct region and non-infarct region in this type of staining is the main obstacle for locating the infarcted area [27]. As H&E staining is not commonly used to qualitatively understand the morphology of a myocardial infarct, and the border of the infarct region is not readily distinguishable from other tissue regions, this limits the use of H&E staining in assessments of myocardial infarct. However, with our simple algorithm, the infarct region is revealed as a notably and visibly distinct border (Figure 2B and Figure 4B). To our knowledge, no previous approach has been as successful in bringing infarcted regions in H&E-stained tissues into such definitive contrast with non-infarcted regions. Overall, successful detection and isolation of the infarct region in H&E-stained cardiac tissue provides a new approach to assess the severity of myocardial infarction.

Slightly different from MTS, which stains the tissues with three colors, TTC and PSR, respectively, stain the infarct region with one color and other regions with another color. The distinct difference in color means that our algorithm is able to readily isolate the infarcted region for quantification purposes. Although TTC staining is used to stain living cardiac and brain tissues [25,28], while PSR is used to stain fixed tissues [29,30], our algorithm can effectively detect the border of the infarcted region. Importantly, the algorithm can provide a highly efficient method for quantifying infarct size, especially when processing bulk tissue samples [28,31].

Although our algorithm is designed to assess myocardial infarction, it is potentially able to evaluate the distribution and pattern of sparse fibrotic elements. The sparse and randomly distributed fibrosis in diseased cardiac tissue can make it challenging and tedious to quantify levels of fibrosis using manual tracing [32,33,34]. However, our algorithm provides the advantageous feature of isolating and quantifying such sparse and broadly scattered elements from the entire tissue section, so that the fibrosis level can be qualitatively and quantitively evaluated.

## 5. Limitations

Our algorithm has several limitations. First, the algorithm we provided was developed based on the color differences between the myocardial infarct regions and other regions; therefore, the effectiveness of the algorithm can be affected by staining quality. If the color of the myocardial infarct region is minimally distinct from other regions in the MTS, TTC or PSR-stained sections, the accuracy of infarct quantification can be attenuated. A specific limitation of H&E staining is that it is challenging to evaluate fibrosis levels when the fibrotic tissue is sparse and randomly distributed. As shown in Figure 1, the non-infarcted region still has a strong but sparse green-colored area after the infarct region is revealed. Therefore, our algorithm may not be suitable to investigate the sparse fibrosis in H&E-stained cardiac tissues. Second, this study does not include any experimental groups of non-diseased and diseased hearts with and without any treatment to investigate the efficiency of our algorithm in comparatively evaluating myocardial infarct size. That being said, since our algorithm is highly dependent on the color difference between the infarct region and non-infarct region, we believe our algorithm will provide a reliable tool for evaluating relative comparisons of myocardial infarct volumes between treatments. Lastly, while our algorithm is specifically designed to quantify infarct size in stained cardiac tissues, it is not applicable for evaluating the infarct using non-invasive imaging techniques, such as cardiac MRI or echocardiography, which are typically used to guide treatment decisions. However, by optimizing the therapeutic treatment in preclinical models using our quantification approach, we can contribute to improving treatment planning in clinical settings.

## 6. Conclusions

In conclusion, we successfully developed a simple and effective color-based algorithm to detect, isolate and quantify myocardial infarct size. Notably, this color-based algorithm is also capable of detecting and isolating difficult to resolve myocardial infarct regions in H&E-stained cardiac tissues and proved potentially useful in the evaluation of patchy and diffuse fibrosis in cardiac tissues.

## Figures and Tables

**Figure 1 diagnostics-14-02325-f001:**
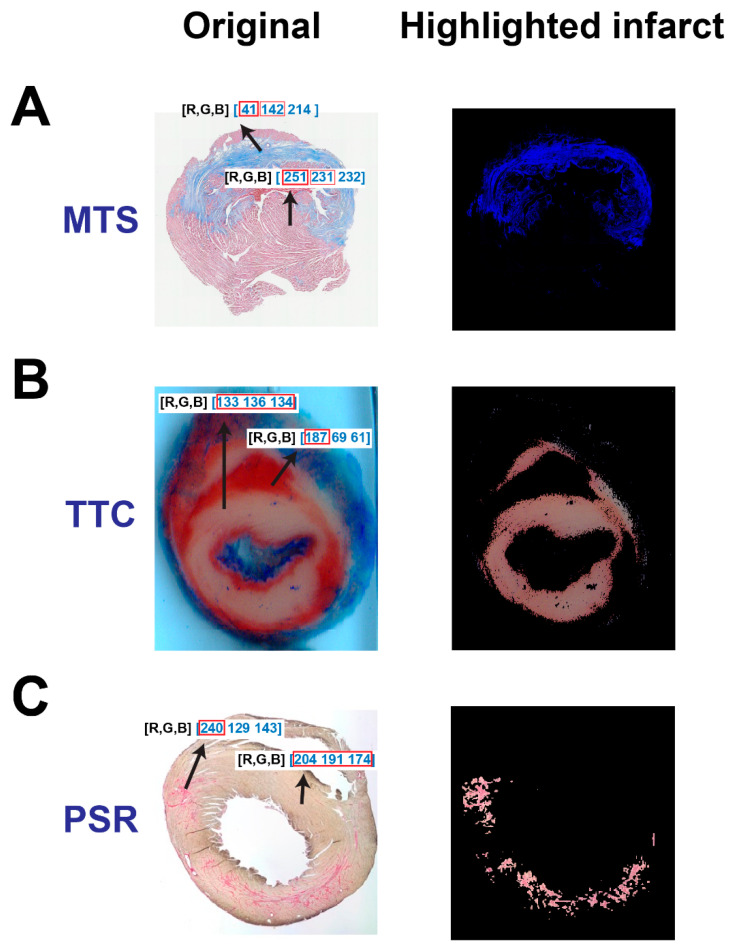
The rationale of the color-based algorithm for detecting and isolating myocardial infarctions from three different histological staining techniques. The original section image and the highlighted infarct after staining with MTS (**A**), TTC (**B**) and PSR (**C**), respectively.

**Figure 2 diagnostics-14-02325-f002:**
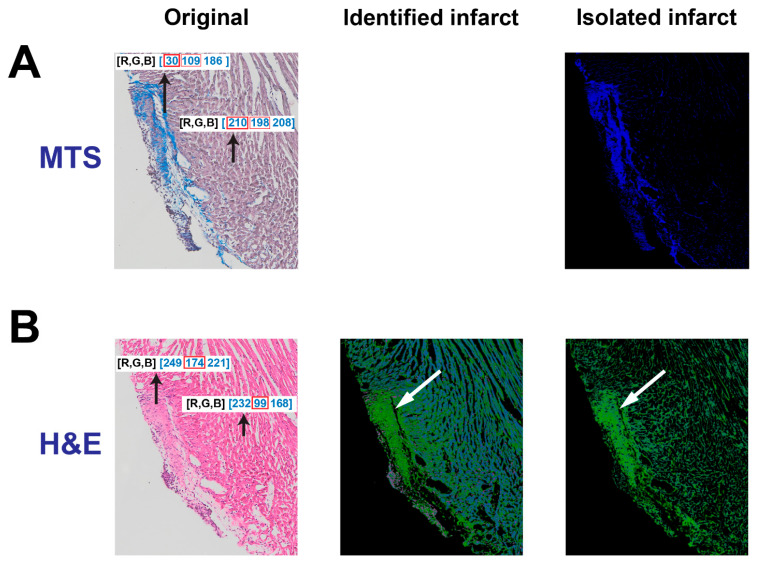
Identification of the infarct region in an H&E-stained tissue section. (**A**) The original tissue and isolated infarct in a MTS-stained tissue section. (**B**) The original tissue, identified infarct and isolated infarct in a H&E-stained tissue section. The black arrow points to a pixel which is composed of three values for RGB respectively. The while arrow points to the infarct location.

**Figure 3 diagnostics-14-02325-f003:**
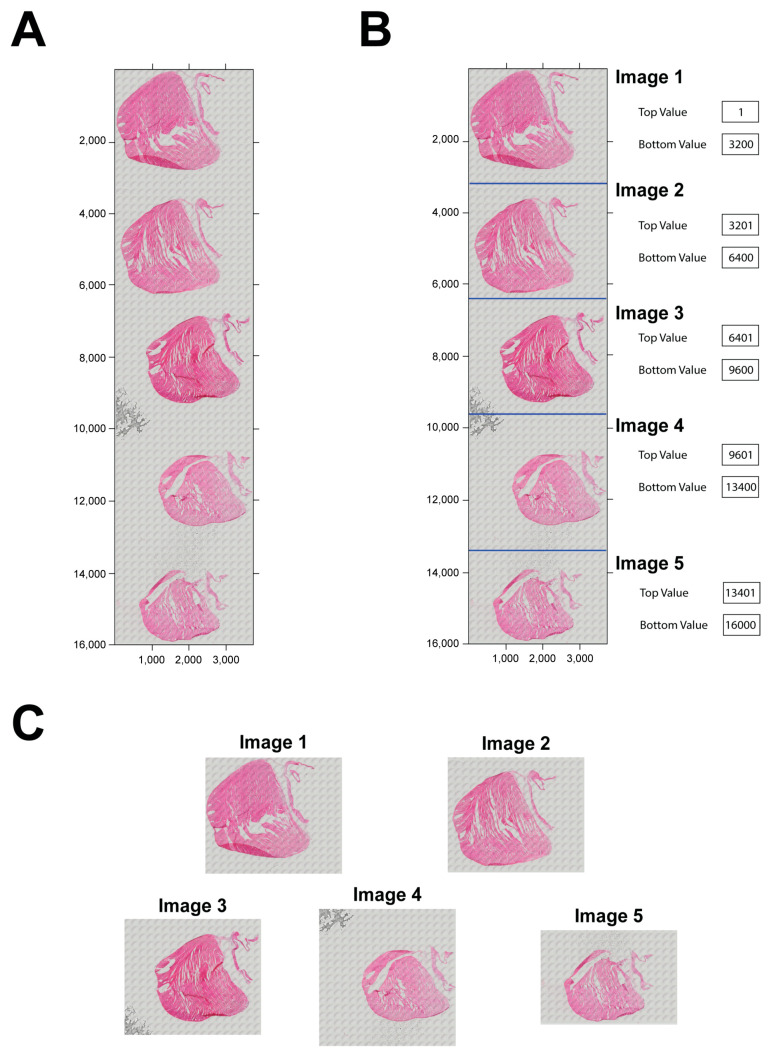
Image splitting. (**A**) The slide image with five tissue sections stained with H&E after scanning with a microscope. (**B**) The tissue sections separated by lines for splitting in the program. The top value and bottom value on the right are used to determine each image. (**C**) The split five images.

**Figure 4 diagnostics-14-02325-f004:**
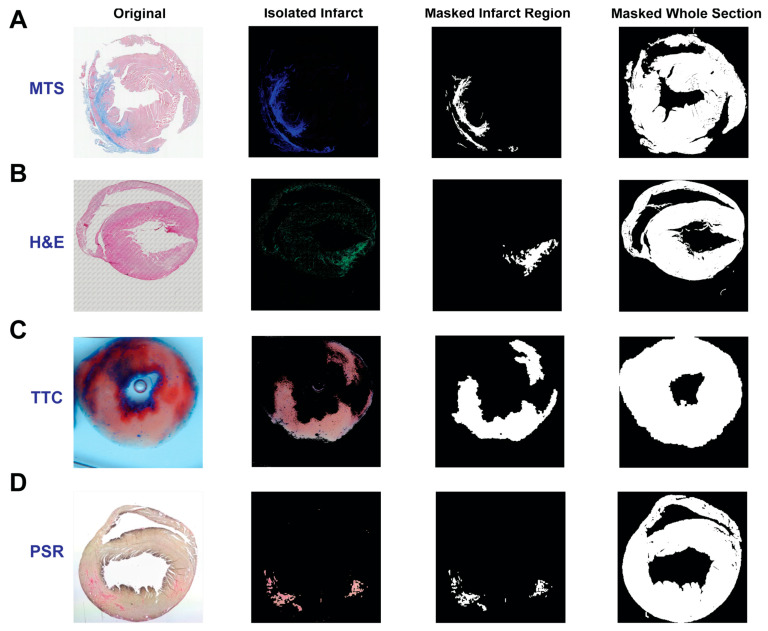
Isolated and masked infarct regions. The infarct region is isolated and masked after staining with MTS (**A**), H&E (**B**), TTC (**C**) and PSR (**D**), respectively. The whole section is masked as well for further quantification.

**Figure 7 diagnostics-14-02325-f007:**
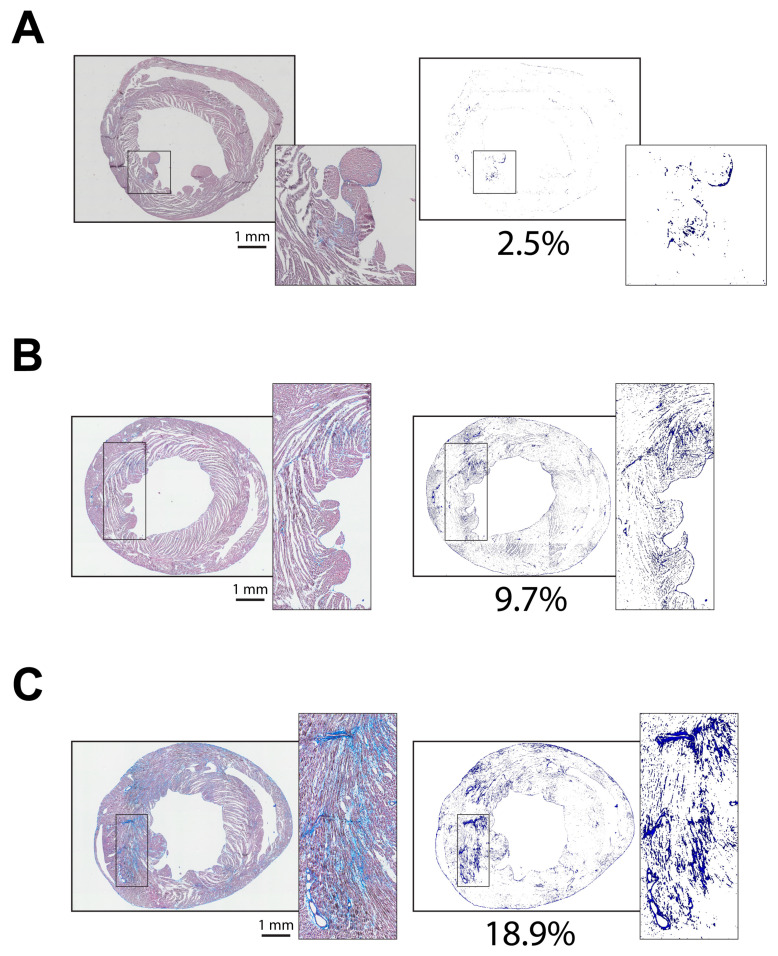
Isolation of diffuse fibrosis from the MTS-stained cardiac sections from three different types of mice. (**A**) The cardiac section and the isolated fibrotic tissues from a 10-week-old wild-type normal mouse heart. (**B**) The cardiac section and the isolated fibrosis from a 10-week-old desmoplakin-deficient mouse heart. (**C**) The cardiac section and the isolated fibrosis from a 22-week-old desmoplakin-deficient mouse heart. The percentage in each panel represents the percentage of fibrosis in the whole section. The magnification for scanning each section is 20×.

## Data Availability

The original contributions presented in the study are included in the article/Appendix A, further inquiries can be directed to the corresponding author.

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
