# Peer review of "Detection, Isolation and Quantification of Myocardial Infarct with Four Different Histological Staining Techniques"

_diagnostics, 2024, doi:10.3390/diagnostics14202325_

Round 1

Reviewer 1 Report

Comments and Suggestions for Authors

This manuscript has demonstrated the segmentation as well as area and volume measurement of the myocardial infarct. The segmentation, area, and volume measurement results have been shown in the manuscript.

1. In the abstract and introduction sections, the authors mentioned that quantification of the myocardial infarct is an important task for therapeutic strategies. Perhaps the authors could elaborate in detail what are among therapeutic follow-up can be done to the patients according to these myocardial infarct quantification results in the abstract, discussion, as well as conclusion section.

Comments on the Quality of English Language

-

Author Response

  1. In the abstract and introduction sections, the authors mentioned that quantification of the myocardial infarct is an important task for therapeutic strategies. Perhaps the authors could elaborate in detail what are among therapeutic follow-up can be done to the patients according to these myocardial infarct quantification results in the abstract, discussion, as well as conclusion section.

We appreciate the reviewer’s suggestion to elaborate on therapeutic strategies based on myocardial infarction (MI) quantification. We would like to clarify that our study focuses on quantifying MI in preclinical models using histological techniques, which are not applicable for direct therapeutic follow-up in human patients. In clinical settings, MI size is typically assessed through non-invasive imaging modalities such as cardiac MRI or echocardiography, which can guide treatment decisions. However, histological quantification of MI, as done in our study, plays an important role in understanding the extent of myocardial damage in research contexts and informs the development of therapeutic strategies. In order to address the reviewer’s concern, we have revised the discussion section to clarify the limitation of our quantification method of MI. The updated text is provided here as well:

Lastly, while our algorithm is specifically designed to quantify infarct size in stained cardiac tissues, it is not applicable for evaluating the infarct using non-invasive imaging techniques such as cardiac MRI or echocardiography, which are typically used to guide treatment decisions. However, by optimizing the therapeutic treatment in preclinical models using our quantification approach, we can contribute to improving treatment planning in clinical settings.

We hope this revision addresses the reviewer’s concern.

Reviewer 2 Report

Comments and Suggestions for Authors

Myocardial infarction is an important medical and social problem. Assessment of myocardial infarction severity and lesion volume is an urgent task. The article describes a new method of identifying the area of myocardial infarction in histological examination.

Comments:

1.      It is recommended to add a scheme with the design of the study to better understand the time frame from the onset of myocardial infarction to evaluate the myocardial infarction. Based on the description in the materials and methods in the models, there were different time frames from the onset of infarction. The timing is of great clinical importance.

2. The description of materials and methods is very brief. It is recommended that details be added to the study description. How many mice were there in each group with different staining options? It is recommended to describe in more detail the staining procedures, reagents, description of the equipment used for the study.

Author Response

  1. It is recommended to add a scheme with the design of the study to better understand the time frame from the onset of myocardial infarction to evaluate the myocardial infarction. Based on the description in the materials and methods in the models, there were different time frames from the onset of infarction. The timing is of great clinical importance.

We appreciate the reviewer’s suggestion to include a scheme depicting the study design and time frame for myocardial infarction evaluation. However, we would like to clarify that our manuscript is a method-focused study. We developed and validated an algorithm for quantifying myocardial infarction using four different staining techniques, and our analysis was performed on pre-existing cardiac tissue samples, which already had established infarcts. As such, we did not conduct any experiments related to the induction of myocardial infarction or track its progression over time, and thus we do not have a specific study design or time frame to present.

Our work is intended to provide a robust tool for MI quantification, independent of the time points at which infarction occurs. We believe that our method can be widely applied in studies that evaluate myocardial infarction at various stages, but the timing of infarction and subsequent evaluation would depend on the individual experimental or clinical context, which was not the focus of our study.

Additionally, we have begun testing our algorithm in several different projects involving myocardial infarction and fibrosis. Indeed, the posting of the pre-print of our paper to the BioRxiv prompted colleagues from overseas to reach out to us to request help with our novel method and algorithms. We view that it is important to provide our software free of charge and support it in the spirit of collaboration and will continue to do so.  In a pilot study, our algorithm can clearly identify the myocardial infarct in HE-stained cardiac tissue and separate the fibrosis from the normal tissues.

We hope this clarifies the scope of our work, and we are happy to revise the manuscript to make this distinction clearer if needed.

  1. The description of materials and methods is very brief. It is recommended that details be added to the study description. How many mice were there in each group with different staining options? It is recommended to describe in more detail the staining procedures, reagents, description of the equipment used for the study.

Thank you very much for the valuable feedback. As mentioned in response to the previous comment, this is a method-focused study, so we didn’t provide very detailed protocols in this study. In order to address the reviewer’s concerns, I have revised and expanded the description of the materials and methods section, which can be found in blue within the manuscript. We hope this additional information resolves the reviewer’s concerns.